# Complete Right Bundle Branch Block as a Predictor of Cardiovascular Events in Type 2 Diabetes

**DOI:** 10.3390/jcm11154618

**Published:** 2022-08-08

**Authors:** Katsuhiro Ono, Sadahiko Uchimoto, Masamune Miyazaki, Natsuki Honda, Katsuhito Mori, Tomoaki Morioka, Takumi Imai, Tetsuo Shoji, Masanori Emoto

**Affiliations:** 1Department of Internal Medicine, Fujiidera Municipal Hospital, Fujiidera 583-0012, Japan; 2Department of Metabolism, Endocrinology and Molecular Medicine, Osaka Metropolitan University Graduate School of Medicine, Osaka 545-8585, Japan; 3Department of Nephrology, Osaka Metropolitan University Graduate School of Medicine, Osaka 545-8585, Japan; 4Department of Medical Statistics, Osaka Metropolitan University Graduate School of Medicine, Osaka 545-8585, Japan; 5Department of Vascular Medicine, Osaka Metropolitan University Graduate School of Medicine, Osaka 545-8585, Japan; 6Vascular Science Center for Translational Research, Osaka Metropolitan University Graduate School of Medicine, Osaka 545-8585, Japan

**Keywords:** bundle branch block, cardiovascular diseases, type 2 diabetes mellitus

## Abstract

Complete right bundle branch block (CRBBB) is generally regarded as a clinically insignificant abnormality on an electrocardiogram, although its predictive value for cardiovascular events in type 2 diabetes mellitus (T2DM) is unknown. We examined the association of CRBBB with cardiovascular events during a 6-year follow-up in a single-center cohort study. The Fine–Gray model was used to analyze the independent association between CRBBB and composite cardiovascular events including cardiovascular death, nonfatal myocardial infarction, nonfatal stroke, and hospitalization for heart failure during follow up. We analyzed the data of 370 T2DM patients including 62 patients with pre-existing heart disease. CRBBB was found in 34 patients (9.2%). The composite cardiovascular outcome was recorded in 32 patients. When analyzed with the Fine–Gray model with inverse probability of treatment weighting, CRBBB was significantly associated with a higher risk of the cardiovascular outcome (hazard ratio, 2.55; 95% confidence interval, 1.04 to 6.26; *p* = 0.041). This association remained significant even after further adjustment for each of the potential confounders. This study suggested that CRBBB was an independent predictor of cardiovascular events in T2DM. Further studies with a larger sample size are warranted.

## 1. Introduction

Complete right bundle branch block (CRBBB) on an electrocardiogram, unlike left bundle branch block, has been generally regarded as a clinically insignificant finding in otherwise healthy individuals [1,2,3,4,5]. In contrast, some studies reported the significant association between CRBBB and cardiovascular events and mortality in patients with heart disease [6,7] and in people without cardiovascular disease [8,9]. These studies suggest that the prognosis of individuals with CRBBB may vary depending on the population. Such association has not been examined in patients with type 2 diabetes mellitus (T2DM). Since patients with T2DM are known to have a higher risk of both CRBBB [10] and cardiovascular events [11], we hypothesized that CRBBB could be predictive of cardiovascular events in patients with T2DM. 

## 2. Materials and Methods

### 2.1. Study Design and Population

This is a single-center cohort study in prevalent T2DM patients. The inclusion criteria were (1) T2DM patients who were treated at the Department of Internal Medicine, Fujiidera Municipal Hospital, Fujiidera, Osaka, Japan on 1 January 2015; (2) patients who were examined with 12-lead electrocardiogram in 2014; and (3) patients whose clinical outcomes during the follow up until 1 January 2021 were available. No specific exclusion criteria were set. 

### 2.2. Exposure and Outcome

The key exposure was the presence versus absence of CRBBB on the latest 12-lead electrocardiogram at rest taken in 2014. CRBBB was defined by the Minnesota Code criteria of a limb lead QRS duration of 0.12 s or more with an R’ greater than R or an R peak duration of 0.06 s or more in either lead V1 or V2 [1].

The outcome was composite cardiovascular events consisting of cardiovascular death (including fatal myocardial infarction, fatal stroke, death from heart failure, and sudden cardiac death), nonfatal myocardial infarction, nonfatal stroke, or hospitalization for heart failure during follow up from 1 January 2015 until 1 January 2021, collected by reviewing medical records. 

### 2.3. Other Variables

Based on many previous studies, we collected data regarding age, sex, smoking, hypertension, dyslipidemia, pre-existing heart disease, body mass index (BMI), estimated glomerular filtration rate (eGFR), and albuminuria as potential confounders. The pre-existing heart disease included coronary heart disease, cardiomyopathy, left ventricular hypertrophy, valvular heart disease, left ventricular diastolic disorder, congenital heart disease, and atrial fibrillation. The duration of T2DM and HbA1c were also recorded as variables related to T2DM. 

Smoking denotes current smoking. Hypertension was defined as systolic blood pressure ≥140 mmHg, diastolic blood pressure ≥90 mmHg, or use of any antihypertensive agent [12]. Dyslipidemia was defined as low-density lipoprotein cholesterol ≥120 mg/dL, triglyceride ≥150 mg/dL, high-density lipoprotein cholesterol <40 mg/dL, or use of any lipid-lowering medication [13]. BMI was calculated by body weight (kg) divided by squared height (m^2^). eGFR was calculated by the equation for the Japanese [14]. Albuminuria was categorized into normo-albuminuria, micro-albuminuria, and overt albuminuria [15]. Duration of T2DM and HbA1c were based on medical records.

### 2.4. Statistical Analysis

The baseline characteristics of the patients were summarized by number (%) for categorical variables and median (interquartile range) for continuous variables. Comparison between the two groups with and without CRBBB was done by Fisher’s exact test or Mann–Whitney’s U-test.

To investigate the association between CRBBB and the cardiovascular outcome, cumulative incidence functions (CIFs) for cardiovascular outcome were estimated using the Gray method, and incidence rates were compared between two groups with and without CRBBB using hazard ratio (HR) and 95% confidence interval (95% CI) from Fine–Gray analysis. Noncardiovascular death was considered as competing risk. To address the problem of confounding by the difference in background characteristics of groups, a propensity score approach was used. Propensity scores were estimated by logistic regression using the following background characteristics: age, sex, hypertension, dyslipidemia, smoking, pre-existing heart disease, eGFR, albuminuria, BMI, HbA1c, and duration of T2DM. The inverse probability of treatment weights (IPTWs) were then calculated based on the estimated propensity scores. The balances of patient background characteristics in the IPTW analysis were assessed by absolute standardized differences (ASDs). ASDs less than 0.1 were considered as balanced between groups [16]. IPTW-adjusted CIF and HR were estimated. The proportional hazard assumption for CRBBB on composite cardiovascular events was assessed by plotting the log of negative log(1−CIF).

Stratified analysis was conducted to examine whether some subgroups of patients were at higher or lower risk of cardiovascular events associated with CRBBB. Subgroups were stratified by sex, median age, the presence of hypertension, the presence of dyslipidemia, smoking status, eGFR (≥60, <60 mL/min/1.73 m^2^), the presence of albuminuria (≥30 mg/g creatinine), the presence of pre-existing heart disease, median HbA1c, and median duration of diabetes.

These statistical calculations were performed using JMP (version 14.3.0, SAS Institute, Tokyo, Japan), EZR (version 1.53), R (version 4.2.0), and SAS (version 9.4). EZR is a graphical interface for R (version 4.0.2, The R Foundation for Statistical Computing, Vienna, Austria) developed by Dr. Kanda, Saitama Medical Center, Jichi Medical University, Saitama, Japan [17].

## 3. Results

We analyzed data from 370 T2DM patients including 167 women (45.1%); median (interquartile range) of age, 71 (64 to 77) years. Among them, 62 patients had pre-existing heart disease. CRBBB was found in 34 out of the 370 patients (9.2%). As shown in Table 1, the group with CRBBB had a higher proportion of men, higher age, and lower eGFR, whereas the prevalence of pre-existing heart disease was not different between the two groups. As shown by Figure 1, the proportion of patients with CRBBB was higher in men and in higher age groups.

During a median follow up of 6.0 years, composite cardiovascular events were recorded in 32 patients out of the total subjects. In the CRBBB (–) group, 24 patients experienced cardiovascular events (12 with cardiovascular death, 4 with nonfatal myocardial infarction, 3 with nonfatal stroke, and 5 with hospitalization for heart failure). In the CRBBB (+) group, eight patients experienced cardiovascular events (two with cardiovascular death, no one with nonfatal myocardial infarction, four with nonfatal stroke, and two with hospitalization for heart failure).

The crude association of CRBBB with the composite cardiovascular events was significant (Figure 2A, Table 2). This positive association remained significant after adjustment with IPTW (Figure 2B, Table 2). There was no clear violation of proportional hazard assumption for CRBBB on composite cardiovascular events based on the log of negative log(1−CIF) plot (Figure 2C). While the balances in patient background characteristics were generally improved by the IPTW method, small imbalances in age, eGFR, BMI, HbA1c, and pre-existing heart disease remained (Figure 2D). Therefore, we performed additional Fine–Gray analysis adjusting these characteristics in the model. The positive association remained significant even after this additional adjustment (Table 2).

The results of stratified analysis are shown in Figure 3. The positive association between CRBBB and subsequent CVD events was not evident in some subgroups, particularly in women, those of older age (≥71 years), those with preserved kidney function, and those without albuminuria. In addition, the association was not significant in those without vascular risk factors such as hypertension, dyslipidemia, smoking, high HbA1c, and long duration of diabetes.

## 4. Discussion

This study is the first that reported CRBBB as an independent risk factor of cardiovascular events in a cohort of T2DM. This finding in T2DM may be important, since CRBBB is generally perceived as clinically insignificant [1,2,3,4,5].

Significant association of CRBBB with worse prognosis was reported by meta-analyses in patients with acute myocardial infarction [6,7] and patients with heart failure [6]. Of note, at least two large cohort studies revealed that CRBBB was associated with higher risk of cardiovascular disease and all-cause mortality in patients without known cardiovascular disease [9] and in the general population [8]. Thus, the degree of risk associated with CRBBB appears to vary depending on the populations. This study has added T2DM to the list of patient groups in which CRBBB predicts worse outcomes.

In our study, the association between CRBBB and cardiovascular risk in T2DM was significant in statistical models in which the effects of pre-existing heart disease and other potential confounders were considered, and the hazard ratios were similar between subgroups with and without pre-existing heart disease. These results suggest that the risk of cardiovascular events was elevated in individuals with CRBBB among T2DM patients regardless of pre-existing heart disease. Diabetic cardiomyopathy [18] is the concept that the myocardium is impaired by hyperglycemia, insulin resistance, advanced glycation end product accumulation, lipo-toxicity, activated renin–angiotensin–aldosterone system, oxidative stress, and other mechanisms. CRBBB in T2DM patients may be a sign of undiagnosed diabetic cardiomyopathy, which could be followed by clinically significant cardiovascular outcomes.

Stratified analysis indicated that the association between CRBBB and risk for subsequent CVD events was not evident in some subgroups such as women, those with older age, those without decreased eGFR, those without albuminuria, and those without vascular risk factors. These subgroups, excluding those with older age, are generally regarded to have a lower risk of CVD. It may be that CRBBB increases the risk of CVD in T2DM patients when vascular risk factors are present regardless of pre-existing heart disease and that such association is less evident in older patients.

This study has limitations including a retrospective design, a single-center observation in Japan, and its small sample size, whereas its strength includes careful statistical analysis using competing risk models and adjustment with IPTW.

## 5. Conclusions

In conclusion, this study suggests that CRBBB is an independent predictor of cardiovascular events in T2DM. Due to the small sample size of this cohort, further studies are needed to confirm and establish the predictive value of CRBBB in this population.

## Figures and Tables

**Figure 1 jcm-11-04618-f001:**
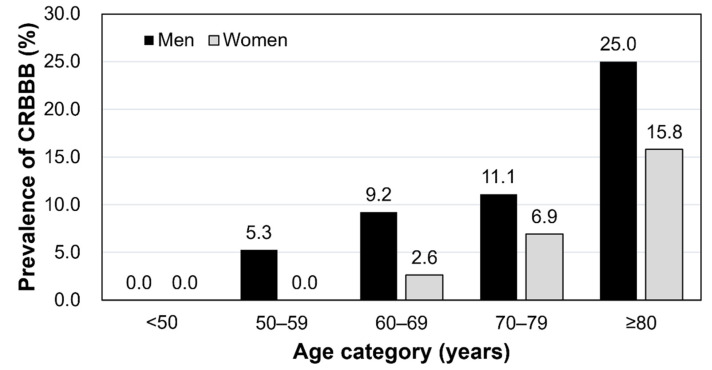
**Prevalence of CRBBB by sex and age category.** The bar graph shows prevalence (percentage) of CRBBB by age and sex. Abbreviation: CRBBB, complete right bundle branch block.

**Figure 2 jcm-11-04618-f002:**
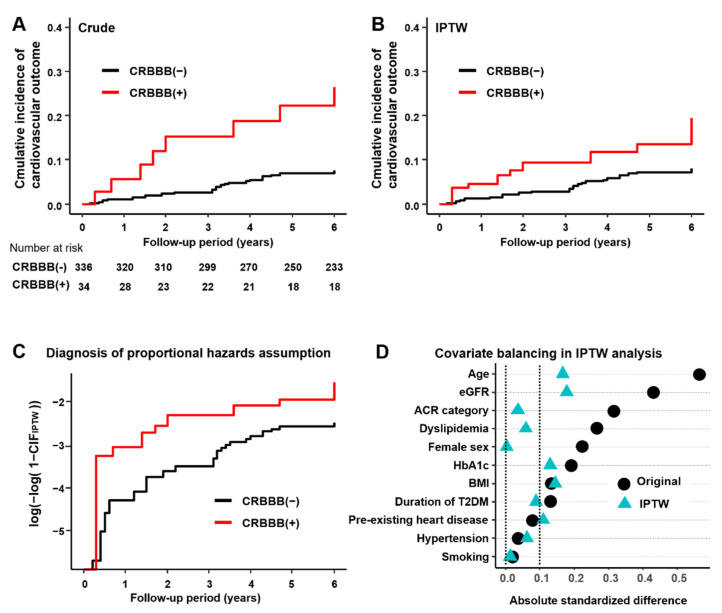
Association of CRBBB with cardiovascular events. (**A**) Crude cumulative incidence of the composite cardiovascular outcome with noncardiovascular death as a competing risk. (**B**) Cumulative incidence of the composite cardiovascular outcome with noncardiovascular death as a competing risk estimated by IPTW method. (**C**) Log of negative log(1−CIF) plot for the diagnosis of proportional hazards assumption of cumulative incidence estimated by IPTW method. (**D**) Assessment of covariate balancing in analysis by IPTW method. Abbreviations: CIF, cumulative incidence function; CRBBB, complete right bundle branch block.

**Figure 3 jcm-11-04618-f003:**
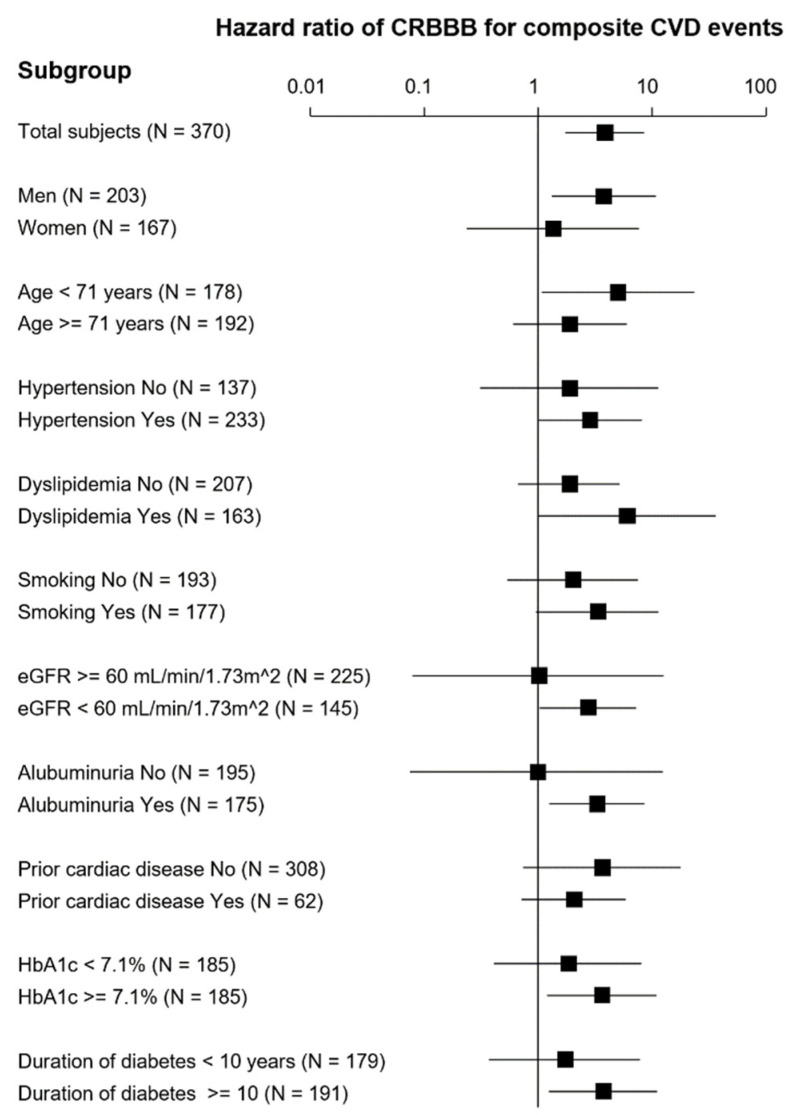
**Stratified analysis of the association of CRBBB with cardiovascular events.** The figure gives hazard ratios and 95% confidence intervals for each group by IPTW-adjusted Fine–Gray model.

**Table 1 jcm-11-04618-t001:** Characteristics of T2DM patients by CRBBB.

Characteristics	Total	CRBBB (–)	CRBBB (+)	*p* Value
Number	370	336	34	---
Age (years)	71 (64–77)	71 (64–76)	75 (69–81)	0.004
Female sex *N* (%)	167 (45.1%)	155 (46.1%)	12 (35.3%)	0.226
Pre-existing heart disease *N* (%)	76 (20.5%)	68 (20.2%)	8 (23.5%)	0.658
Coronary heart disease *N* (%)	28 (7.6%)	27 (8.0%)	1 (2.9%)	0.495
Cardiomyopathy *N* (%)	9 (2.4%)	8 (2.4%)	1 (2.9%)	0.584
Left ventricular hypertrophy *N* (%)	6 (1.6%)	5 (1.5%)	1 (2.9%)	0.442
Valvular heart disease *N* (%)	14 (3.8%)	12 (3.6%)	2 (5.9%)	0.375
Left ventricular diastolic disorder *N* (%)	3 (0.8%)	3 (0.9%)	0 (0.0%)	1.000
Congenital heart disease *N* (%)	3 (0.8%)	3 (0.9%)	0 (0.0%	1.000
Atrial fibrillation *N* (%)	27 (7.3%)	22 (6.6%)	5 (14.7%)	0.089
Current smoker *N* (%)	177 (47.8%)	161 (47.9%)	16 (47.1%)	0.924
Hypertension *N* (%)	233 (63.0%)	211 (62.8%)	22 (64.7%)	0.826
Dyslipidemia *N* (%)	163 (44.1%)	152 (45.2%)	11 (32.4%)	0.149
eGFR (mL/min/1.73 m^2^)	65.0 (54.7–77.5)	65.4 (55.5–77.8)	59.7 (45.4–67.2)	0.017
Albuminuria (normo/micro/overt) *N* (%)	195/110/65 (52.7%/29.7%/17.6%)	180/101/55 (53.6%/30.1%/16.4%)	15/9/10 (44.1%/26.5%/29.4%)	0.161
BMI (kg/m^2^)	23.8 (21.3–26.9)	23.9 (21.3–26.9)	22.7 (21.2–26.9)	0.343
HbA1c (%)	7.1 (6.5–8.1)	7.1 (6.5–8.1)	6.9 (6.3–8.1)	0.206
Duration of T2DM	10 (5–16)	10 (4–16)	9 (5–22)	0.628
SBP (mmHg)	138 (124–149)	138 (124–150)	137 (127–143)	0.347
DBP (mmHg)	75 (66–84)	75 (67–84)	72 (65–77)	0.035
TC (mg/dL)	190 (165–213)	191 (165–213)	185 (158–207)	0.340
LDL-C (mg/dL)	110 (91–131)	110 (91–132)	110 (80–129)	0.488
HDL-C (mg/dL)	55 (44–66)	55 (45–66)	52 (41–66)	0.713
Triglyceride (mg/dL)	115 (78–162)	115 (78–166)	113 (89–142)	0.669
** *Medications* **				
Anticoagulants	25 (6.8%)	21 (6.3%)	4 (11.8%)	0.222
RASi *N* (%)	186 (50.3%)	169 (50.3%)	17 (50.0%)	0.974
CCB *N* (%)	155 (41.9%)	145 (43.2%)	10 (29.4%)	0.122
β-blocker *N* (%)	36 (9.7%)	35 (10.4%)	1 (2.9%)	0.161
MRA *N* (%)	15 (4.1%)	13 (3.9%)	2 (5.9%)	0.571
Statin *N* (%)	159 (43.0%)	146 (43.5%)	13 (38.2%)	0.558
Insulin *N* (%)	73 (19.7%)	68 (20.2%)	5 (14.7%)	0.440
SGLT2i *N* (%)	5 (1.4%)	5 (1.5%)	0 (0%)	0.474
GLP-1 RA *N* (%)	2 (0.5%)	2 (0.6%)	0 (0%)	0.652
Biguanide *N* (%)	89 (24.1%)	85 (25.3%)	4 (11.8%)	0.079
DPP4i *N* (%)	229 (61.9%)	210 (62.5%)	19 (55.9%)	0.449
Sulfonylurea *N* (%)	139 (37.6%)	130 (38.7%)	9 (26.5%)	0.161
Thiazolidine *N* (%)	31 (8.4%)	30 (8.93%)	1 (2.94%)	0.230
Glinide *N* (%)	5 (1.4%)	4 (1.20%)	1 (2.94%)	0.400
α-GI *N* (%)	56 (15.1%)	52 (15.5%)	4 (11.8%)	0.565

The table gives number (%) for categorical variables and median (interquartile range) for continuous variables. *p* values by Mann–Whitney’s U test and Pearson’s chi-squared test are indicated. Abbreviations: T2DM, type 2 diabetes mellitus; CRBBB, complete right bundle branch block; eGFR, estimated glomerular filtration rate; BMI, body mass index; HbA1c, glycohemoglobin A1c; SBP, systolic blood pressure; DBP, diastolic blood pressure; TC, total cholesterol; LDL-C, low-density lipoprotein cholesterol; HDL-C, high-density lipoprotein cholesterol; RASi, renin–angiotensin system inhibitor; CCB, calcium channel blocker; MRA, mineralocorticoid receptor antagonist; SGLT2i, sodium-glucose transporter 2 inhibitor; GLP-1 RA, glucagon-like peptide-1 receptor antagonist; DPP4i, dipeptidyl peptidase 4 inhibitor; α-GI, α-glucosidase inhibitor.

**Table 2 jcm-11-04618-t002:** Unadjusted and adjusted association of CRBBB with cardiovascular events in T2DM.

		CRBBB (–)	CRBBB (+)	
Number of patients		312	26	
Number of cases		24	8	
Patient-years		1598	106	
Crude rate per 1000 patient-years		16.3	75.6	
Model	Adjustment	HR (95% CI) by Fine–Gray model		*p* value
1	Crude	1.00 (Reference)	3.85 (1.74–8.53)	0.001
2	IPTW method	1.00 (Reference)	2.55 (1.04–6.26)	0.041
3	IPTW method + covariates adjustment *	1.00 (Reference)	3.05 (1.30–7.13)	0.010

The association of baseline CRBBB with risk of a composite of cardiovascular event during follow up was analyzed with the Fine–Gray model. * In Model 3, the following covariates with absolute standardized difference by IPTW method larger than 0.1 were additionally adjusted in the Fine–Gray model: age, eGFR, HbA1c, BMI, and pre-existing heart disease. Abbreviations: CRBBB, complete right bundle branch block; HR, hazard ratio; 95% CI, 95% confidence interval; T2DM, type 2 diabetes mellitus; BMI, body mass index; eGFR, estimated glomerular filtration rate; HbA1c, glycohemoglobin A1c; and IPTW, inverse probability of treatment weighting.

## Data Availability

The data presented in this study are available on request from the corresponding author. The data are not publicly available due to ethics committee permission.

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
