# Peer review of "Complete Right Bundle Branch Block as a Predictor of Cardiovascular Events in Type 2 Diabetes"

_jcm, 2022, doi:10.3390/jcm11154618_

Round 1

Reviewer 1 Report

Ono et al. wrote a manuscript regarding the association between complete right bundle block and cardiovascular adverse outcomes in patients with diabetes mellitus type 2.

The study sounds interesting however, there are a few comments that need to be addressed:

1. Introduction is poor,  could you explain the context in which your research is integrated? Could you expand the reasons that addressed your research? What could be the link between RBBB and T2DM and its influence on adverse outcomes?

2. Materials and methods are weak, how did you enroll the patients? When? Inclusion and exclusion criteria need to be written, too.

3. Why didn't you include fatal myocardial infarction and fatal stroke in your composite outcome? To be consistent with the results presented, I suggest changing the term "cardiovascular death" (page 2, exposure and outcome) and to use composite cardiovascular events, as in Figure 2/results.

4. In the results, when you report the events, you wrote "cardiovascular death" instead of "sudden cardiac death", as you listed in the paragraph "exposure and outcomes", please correct accordingly.

5. I suggest providing a unique model instead of single-variable models (e.g. model 2 - age and sex - plus the single variables in table 2) in order to increase the value of your work. 

6. It is difficult to draw a conclusion because of the strict number of patients with RBBB (only 34). I suggest caution in the discussion due to the possible biased results. Patients with RBBB were older and mostly men; this suggests that they were at risk of cardiovascular events on their own, regardless of the presence of RBBB.

7. May it possible to have atrial fibrillation listed in the comorbidities? (This may suggest the reason for usage of anticoagulants). Moreover, could you please list the variable "prior cardiac disease"? What does it mean?

8. Revise for typo and grammar errors (e.g. "Inconclusion").

Reviewer 2 Report

Dear Sir/Madam,

I had the opportunity to act as a reviewer on the recent submission by Ono et al. to the Journal of Clinical Medicine.

The authors present very interesting original research studying the complete right bundle branch block as a predictor for cardiovascular events in type 2 diabetes mellitus. The main finding of the study is that the complete right bundle branch block was an independent predictor of cardiovascular events in patients with type 2 diabetes mellitus.

The results are very interesting. However, some issues need to be addressed:

  1. Regarding the statistics used in the study: in order to use the Fine-Gray subdistribution model the predictor variables must fulfill the proportional hazard assumption (i.e., goodness of fit). Otherwise, the method is not suitable. Please provide this critical information in the manuscript.
  2. Why did the authors choose to study the right bundle branch block and not the left bundle branch block?
  3. According to the authors the prognosis of individuals with CRBBB may vary depending on the population (lines 38-39): please provide reference.
  4. Why did the authors choose to study the exact risk factors enumerated on lines 56-59 (i.e., history of major adverse cardiovascular events such as stroke)? What does exactly pre-existing heart disease mean?
  5. It is not clear what Figure 3 means: are these the effects of the predictor variables on the outcome variable? If so, which method was employed? (i.e., LR test)

Minor issues:

1-     Figure 1 has a typo on the ordinate axis (“Prevalend”)

Best regards,

Round 2

Reviewer 2 Report

Dear Sir/Madam,

Thank you for reviewing the manuscript and addressing the mentioned issues.

Please provide in the manuscript the data supporting the “Change L” (the plot).

Best regards
